# E²GRAPHRAG: ADVANCING THE PARETO FRONTIER IN EFFICIENCY AND EFFECTIVENESS FOR GRAPH-BASED RAG

## ABSTRACT

Graph-based RAG methods like GraphRAG demonstrate strong global understanding of the knowledge base by constructing hierarchical entity graphs, but often suffer from inefficiency and rigid, manually defined query modes, limiting practical use. To address these limitations, we present E²GraphRAG, a streamlined graph-based RAG framework that advances the Pareto frontier of Efficiency and Effectiveness. In the indexing stage, E²GraphRAG utilizes large language models to generate a summary tree, and NLP tools to construct an entity graph from document chunks, with bidirectional indexes linking entities and chunks for efficient lookup. In the retrieval stage, the graph structure filters related entities, while the bidirectional indexes map these entities to their corresponding chunks, supporting an adaptive mechanism that dynamically switches between local and global modes. Experiments show that E²GraphRAG achieves up to $10\times$ faster indexing than GraphRAG while maintaining comparable QA performance, advancing the Pareto frontier with respect to effectiveness and efficiency. Our code is available at `https://anonymous.4open.science/r/E-2GraphRAG-8897`.

*"Everything should be made as simple as possible, but not simpler."*

— Albert Einstein

## 1 INTRODUCTION

With the continuous advancement, large language models (LLMs) (Dao et al., 2022; Pope et al., 2023; Vaswani et al., 2017) have become a cornerstone in NLP, which have been widely applied in tasks such as text summarization (Kirstein et al., 2024; Nakshatri et al., 2023), machine translation Koshkin et al. (2024); Lu et al. (2024), and question answering (Chen et al., 2024b; Li et al., 2024b; Schimanski et al., 2024). However, they still face limitations, including hallucinations (Du et al., 2024; Ramprasad et al., 2024; Sahoo et al., 2024; Sriramanan et al., 2024) and a lack of domain-specific knowledge (Jiang et al., 2024; Liu et al., 2024; Shen et al., 2024; Wang et al., 2025b). To address these issues, Retrieval-Augmented Generation (RAG) has been proposed (Fan et al., 2024; Laban et al., 2024; Lewis et al., 2020). By retrieving relevant knowledge from external sources and leveraging the in-context learning capabilities of LLMs, RAG allows models to integrate timely and domain-specific information, thereby mitigating issues such as hallucinations and knowledge gaps.

Traditional RAG methods typically retrieve only a small set of chunks from original documents as supplemental knowledge. However, this limited context could be insufficient for providing the model with a comprehensive and global understanding of the knowledge base, such as understanding and summarizing a character's personality transformation, as in NovelQA (Wang et al., 2025a). Consider the novel *Harry Potter and the Prisoner of Azkaban* and the question: "*Peter Pettigrew used to be positive and finally becomes a negative one. Tell in one sentence what marks this character's change.*" Traditional RAG methods typically retrieve only a few isolated chunks about Peter Pettigrew, whereas answering this question requires a comprehensive understanding of his entire character arc.

To address the problem, existing state-of-the-art methods, including RAPTOR (Sarthi et al., 2024), GraphRAG (Edge et al., 2025), and LightRAG (Guo et al., 2024), adopt an *indexing-and-retrieval*

*paradigm*: they first use LLMs to index the documents into tree- or graph-based structures[1] and then retrieve on these structured data. While hierarchical trees, constructed by recursively merging text chunks, provide global understanding, they are limited in capturing fine-grained knowledge, such as entities and their relations. Entity graphs, on the other hand, enable the extraction and integration of such fine-grained knowledge across dispersed chunks, but they heavily rely on LLM-based entity and relation extraction, leading to substantial computational and time costs during indexing.

To summarize, existing approaches face three major challenges. First, efficiency remains the primary bottleneck: although several efforts (fas, 2024; Guo et al., 2024) have attempted to reduce computational overhead, indexing and retrieval are still far from optimal. Second, most methods rely exclusively on either tree or graph structures to organize raw, lengthy text. While each structure has its advantages, their joint integration has not been thoroughly investigated. Third, in the retrieval stage, some approaches (Edge et al., 2025; Guo et al., 2024) depend on manually pre-defined query modes (e.g., local or global), resulting in limited flexibility. Therefore, a research question naturally emerges: *Is it possible to design a graph-based RAG model that advances the Pareto frontier of efficiency and effectiveness, while adaptively responding to queries at varying granularities?*

In this paper, we streamline graph-based RAG for high efficiency and effectiveness, and propose the $E^2$GraphRAG model, which combines the strengths of both tree and graph structures. Specifically, we first recursively merge and summarize text chunks to construct a **hierarchical tree**, enabling multi-granularity summarization of raw text. To integrate fine-grained knowledge from dispersed chunks, we also construct a concise **entity graph**. Rather than relying on LLMs for entity extraction, we employ the standard NLP tools such as SpaCy (Honnibal et al., 2020), and define relations based on entity co-occurrence within a sentence. We further construct bidirectional entity-to-chunk and chunk-to-entity indexes to bridge the entity graph and the summary tree, facilitating efficient lookup during subsequent retrieval. In the retrieval stage, we introduce a lightweight adaptive strategy that leverages the entity graph to select between local and global query modes: queries whose entities are densely connected are processed locally, while others fall back to global retrieval. This mechanism models structural relationships among entities explicitly, eliminating the need for manually predefined query modes and enabling more efficient and flexible retrieval for diverse query types.

In summary, our contributions are threefold:

- We propose $E^2$GraphRAG, a novel framework that integrates a summary tree and an entity graph via bidirectional indexes, bringing new insights into lightweight graph-based RAG indexing by constructing the entity graph without relying on LLMs.
- We design a graph-driven adaptive retrieval mechanism that automatically switches between local and global modes, eliminating the need for manual query presets.
- We conduct extensive experiments showing that $E^2$GraphRAG achieves up to 10× faster indexing than GraphRAG while maintaining comparable QA performance, advancing the Pareto frontier.

## 2 RELATED WORK

RAG has been extensively studied, where most existing methods fall into two main categories based on the type of external knowledge source. Most approaches (Asai et al., 2024; Yao et al., 2023) rely on unstructured textual knowledge bases, which are easy to organize and adaptable to various tasks, but often lack a global and structured understanding of the content. Others utilize structured knowledge graphs (He et al., 2024; Li et al., 2024a; Sun et al., 2024), which naturally support multi-hop reasoning and information aggregation. However, building high-quality, domain-specific knowledge graphs typically requires substantial expert efforts and is difficult to scale.

GraphRAG (Edge et al., 2025) is the first method to automatically construct knowledge graphs from raw text, supporting both local and global queries, which attracts considerable attention (Peng et al., 2024; Zhou et al., 2025). While it achieves strong effectiveness through multi-granularity reasoning and community-based summarization, its indexing and retrieval incur substantial costs due to numerous LLM calls and complex JSON outputs. To improve efficiency, subsequent methods explored different trade-offs. LightRAG (Guo et al., 2024) and FastGraphRAG (fas, 2024) eliminate community summarization, with LightRAG directly extracting low- and high-level nodes from each

---

[1]Since tree is a special form of graph, we uniformly use graph-based RAG in this paper.

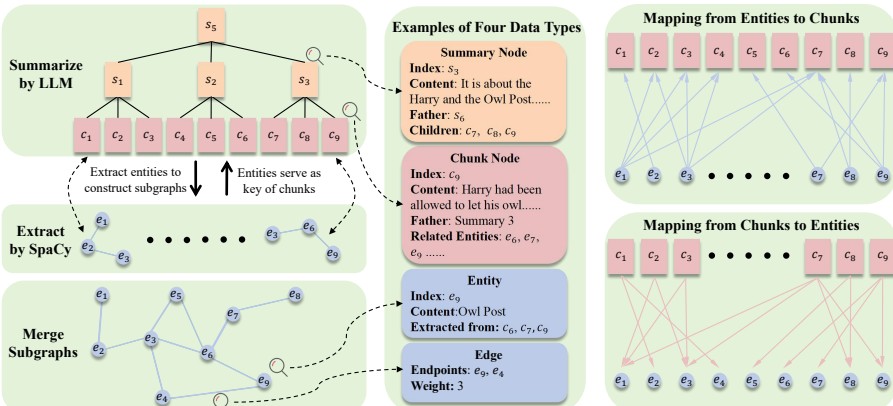

Figure 1: Overview of the indexing stage of E²GraphRAG. The left part shows the indexing tasks, the center presents the four data structures, and the right part displays the two constructed indexes.

chunk, and FastGraphRAG leveraging PageRank for query-time aggregation. However, both methods still rely heavily on LLMs to produce verbose structured outputs. HippoRAG (Gutiérrez et al., 2024) and HippoRAG2 (Gutiérrez et al., 2025) further reduce indexing complexity by extracting only entity and relation names. They employ PageRank to redistribute weights among query-relevant entities to aid retrieval, but at the expense of global reasoning and still depending on LLM-based extraction. LazyGraphRAG defers all LLM calls to retrieval, minimizing indexing cost but introducing high query-time latency. In contrast, RAPTOR adopts a hierarchical summary tree to efficiently preserve multi-level context, offering a lightweight indexing strategy while trading off detailed entity-level reasoning. Overall, existing methods illustrate a clear tension between efficiency and effectiveness: graph-based methods favor performance but incur high computational cost, whereas tree- or simplified-graph methods improve efficiency at the cost of global comprehension or fine-grained knowledge.

Different from the above, E²GraphRAG leverages traditional NLP tools to efficiently construct an entity co-occurrence graph for capturing relationships among entities, while simultaneously building a hierarchical summary tree to preserve multi-granularity information. This design enhances retrieval effectiveness while maintaining high efficiency, resulting in comparable QA performance.

## 3 METHOD

Similar to GraphRAG and other methods, our approach consists of two main stages: **indexing** and **retrieval**. For our task, we first introduce some symbolic definitions to facilitate clearer explanations in subsequent sections. As input, we use $D$ to represent the document, $q$ to denote the query, and $k$ to denote the maximum number of chunks retrieved.

### 3.1 INDEXING STAGE

As in standard RAG indexing, we first split each document into $n$ chunks. We tokenize the document using the tokenizer corresponding to the model used in the subsequent summarization task, and divide it into chunks of 1200 tokens each, with an overlap of 100 tokens between adjacent chunks to mitigate the semantic loss caused by potential sentence fragmentation. The resulting chunked document is denoted as $D = \{c_1, c_2, \cdots, c_n\}$. Then, as illustrated in Figure 1, the indexing stage comprises two main tasks: construction of a **summary tree** and extraction of an **entity graph**. To enhance subsequent retrieval, we further introduce two types of indexes that establish many-to-many mappings between the tree and the graph.

For the summary tree construction, we preserve the original chunk order and employ an LLM to summarize every consecutive group of $g$ chunks. Notably, since most modern LLMs have been extensively trained on text summarization tasks during the instruction tuning (Sanh et al., 2022; Wei et al.), we adopt a minimal prompting strategy—providing only task instructions without lengthy few-shot examples, —thereby improving indexing efficiency. Once all the original chunks have been

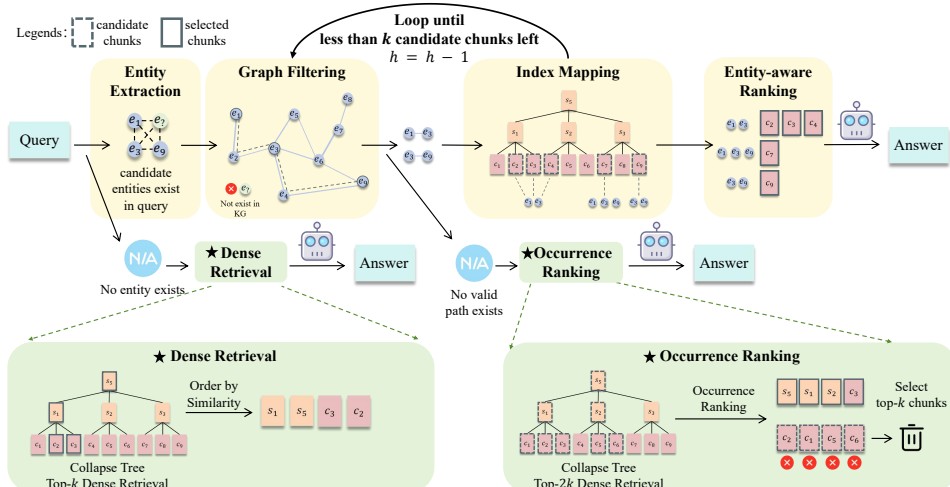

Figure 2: The retrieval stage of E²GraphRAG. Operations belonging to the local retrieval are highlighted in light yellow, while those for global retrieval are highlighted in light green and marked with a ★.

summarized, the resulting summaries are treated as a new sequence of inputs. This recursive summarization process continues, grouping every $g$ summaries at each level, until only $g$ or fewer segments remain. Through the above procedure, the raw document is transformed into a tree structure, where the leaf nodes correspond to chunks and the intermediate or root nodes correspond to the summaries. Nodes closer to the root contain more global and abstract information, while those nearer to the leaves retain more detailed and specific content. We then utilize a pretrained embedding model to encode all chunks and summaries, storing the resulting vectors using *Faiss* (Douze et al., 2024) to enable efficient dense retrieval. Formally, we denote the summary tree as $T = \{c_1, \cdots, c_n, s_1, \cdots, s_o\}$, where each chunk $c_i$ and summary $s_i$ corresponds to a node in the tree.

For the entity graph extraction task, instead of relying on LLMs to extract entities and relations as in GraphRAG-style approaches, we employ lightweight entity extractors, such as SpaCy (Honnibal et al., 2020), NLTK (Bird & Loper, 2004), and fine-tuned BERT (Tjong Kim Sang & De Meulder, 2003), which are significantly more efficient than LLMs for large-scale information extraction. In particular, we extract named entities and common nouns (as nouns often indicate potential entities), and uniformly refer to them as *entities* hereafter. Formally, for each chunk $c_i$, we denoted the extracted entities as $\mathcal{E}_{c_i} = \{e_{c_i}^1, \cdots, e_{c_i}^m\}$, where $m$ is the number of entities identified in chunk $c_i$. After extracting entities, we construct an undirected weighted edge between any two entities that co-occur within the same sentence, where the edge weight reflects their sentence-level co-occurrence frequency. This results in a subgraph $\mathcal{G}_{c_i}$ for each chunk $c_i$, which captures the relations among entities mentioned within the chunk and allows us to construct associations between entities and chunks. To support efficient retrieval, we build two one-to-many indexes to link entities and chunks, thereby capturing the many-to-many relations between them. The entity-to-chunk index, $I_{e \to c}(\cdot)$, maps each entity to the set of chunks where it appears. The chunk-to-entity index, $I_{c \to e}(\cdot)$, records the entities extracted from each chunk. These two indexes establish a many-to-many mapping between the entities in the entity graph and the chunks in the summary tree, facilitating the subsequent entity-aware retrieval stage. For the entire document, we merge all chunk-level subgraphs into a single graph $\mathcal{G}$, where identical entities are unified and edges with the same source and target entities have their weights summed. Since some entities appear in multiple chunks, this merging allows the graph to capture the co-occurrence relationships among entities across the entire document.

In conclusion, as illustrated in Figure 1, our method involves four types of data stored in two data structures: summary nodes and original chunk nodes in the tree, along with entities and weighted edges in the graph. In addition, our method relies on two key indexes, chunk-to-entity index $I_{c \to e}(\cdot)$ and entity-to-chunk index $I_{e \to c}(\cdot)$, which bridge the tree and the graph. These indexes enable efficient mapping from a chunk to its associated entities, and from an entity to the chunks in which it appears, respectively, thereby facilitating subsequent retrieval.

## 3.2 RETRIEVAL STAGE

In the retrieval stage, previous work faces two main challenges: (1) global queries heavily rely on LLMs, resulting in high retrieval latency, and (2) the retrieval hierarchy and methods often require manual specification, introducing additional hyperparameters that are difficult to optimize. To address these issues, we first introduce a novel retrieval mechanism that adaptively selects between global and local retrieval when specific logical conditions are met. Then, we rank and format the retrieved pieces of evidence, therefore enhancing the LLM. To clearly distinguish between the two adaptively selected retrieving modes, we highlight **global retrieval** starting with a ★ throughout this section. The complete pseudo-code is provided in Appendix A, and an overview of our retrieval and ranking pipeline is shown in Figure 2.

At the core of our approach is the intuition that each local query typically involves relevant entities, like "*Slytherin*" and "*House Cup*" in the question "*Has Slytherin won the House Cup?*", and potential relationships among these entities can guide the retrieval process by identifying the most relevant chunks. Therefore, we first use the entity extractor, as in the indexing stage, to extract entities from the query, denoted as $\mathcal{E}_q = \{e_q^1, \cdots e_q^m\}$. The entities in the query are then mapped to the vertices in our constructed graph. For simplicity, entities that cannot be mapped to any graph vertex are treated as invalid and ignored, as they are likely noise introduced by the entity extractor.

★If no entities are identified, we cannot utilize the entities to support meaningful retrieval. In such cases, the query is treated as a global query, and *Dense Retrieval* is performed over the summary tree. Specifically, we adopt a collapsed-tree dense retrieval approach similar to RAPTOR, leveraging the embedding model used in the indexing stage to encode the query. The similarity between the query embedding and these indexed embeddings is then computed to select the top-$k$ most relevant chunks as supplementary information, which are ranked in descending order of similarity.

Otherwise, since the entity extractor lacks the ability to capture semantic relevance, it often fails to identify the core entities aligned with the query intent, resulting in noisy extractions. Simply mapping these entities to the graph is insufficient for filtering out the noise. Therefore, we introduce a *Graph Filtering* step to retain only the core entities for effective retrieval. The underlying heuristic is that truly relevant entities tend to be semantically related and thus connected in the constructed graph. Formally, they should lie within $h$ hops of each other as neighbors. Specifically, we enumerate all pairwise combinations of entities from the query as candidate entity pairs. For each pair, if the two entities are within $h$ hops in the knowledge graph, they are considered semantically related and retained; otherwise, they are discarded as likely irrelevant. The set of selected entity pairs is denoted as $\mathcal{P}_h$. This step is formally defined in Equation 1, where $\text{Dist}_{\mathcal{G}}(\cdot, \cdot)$ returns the hop count of the shortest path between two entities in the graph. If no path exists, it returns infinity. The hyperparameter $h$ controls the strictness of the filtering and can be adaptively adjusted to balance the number of chunks recalled during the following steps.

$$\mathcal{P}_h = \left\{ (e_q^i, e_q^j) \in \mathcal{E}_q \times \mathcal{E}_q \mid i < j, \text{Dist}_{\mathcal{G}}(e_q^i, e_q^j) \leq h \right\} \tag{1}$$

★After this filtering step, if no entity pairs meet the criteria, i.e., there are no fine-grained, interrelated entities in the query, which means their relations cannot be extracted within several local chunks. In such cases, we classify it as a coarse-grained global query as well. This also includes cases where the query contains only a single entity, as there are no pair-wise combinations. However, unlike the previous scenario, entities related to both question and context are still present and can assist in improving chunk selection. To leverage them, we first retrieve the top-$2k$ chunks from the summary tree based on vector similarity as candidate supplementary chunks. We then apply an *Occurrence Ranking* strategy, ranking these candidate chunks according to the frequency of entity occurrences, defined as $w(c_i) = \text{Count}(c_i, \mathcal{E}_q)$. For each candidate summary node, the weight is recursively computed as the sum of the weights of its child nodes, i.e. $w(s_i) = \sum_{c/s \in T_{\text{child}}(s_i)} w(c/s)$, where $c/s$ may refer to either chunk nodes or summary nodes. This recursive weighting naturally assigns higher scores to high-level summary nodes, aligning with the intuition behind global retrieval. Finally, we rank the candidate chunks by their computed weights and select the top-$k$ highest-ranked ones as supplementary information.

If entity pairs exist, this indicates the presence of fine-grained relational entities in the query. In such cases, we perform *Index Mapping*, leveraging the entity-to-chunk index $I_{e \rightarrow c}(\cdot)$ constructed during the indexing stage. Specifically, for each entity pair $(e_q^i, e_q^j)$ in $\mathcal{P}_h$, we map each entity to the corresponding sets of chunks through the index, and then take their intersection to identify the set of

chunks associated with both entities, denoted as $\mathcal{C}_{\text{evidence}}^{(e_q^i, e_q^j)}$. $\mathcal{C}_{\text{evidence}}$, the union of the $C_{\text{evidence}}^{(e_q^i, e_q^j)}$ means all the candidate chunks. Formally, we define the *Index Mapping* operation with Equation 2.

$$\mathcal{C}_{\text{evidence}} = \bigcup_{(e_q^i, e_q^j) \in \mathcal{P}_h} \mathcal{C}_{\text{evidence}}^{(e_q^i, e_q^j)} = \bigcup_{(e_q^i, e_q^j) \in \mathcal{P}_h} \{I_{e \to c}(e_q^i) \cap I_{e \to c}(e_q^j)\} \quad (2)$$

Once the indexes are mapped, if the number of retrieved chunks does not exceed $k$, we directly return them as the final evidence set. Otherwise, we first attempt to reduce the number of chunks by decreasing the hop threshold $h$ step-by-step, as tighter structural constraints help eliminate less relevant neighbors. This continues until either the number of chunks drops below $k$ or the retrieval returns no chunks at all. If the latter occurs (i.e., the retrieval result becomes empty), we revert to the last non-empty result before the drop and apply an *Entity-Aware Ranking* mechanism to select the top-$k$ chunks from it. This ranking is based on multiple structural and statistical signals derived during retrieval. Specifically, we compute two metrics for each candidate chunk: **Entity Coverage Ranking** counts the number of distinct query-related entities present in the chunk. Chunks covering more entities are prioritized as they are not only more likely to be relevant but also tend to contain more comprehensive contextual information. **Entity Occurrence Ranking** ranks the chunks by the total frequency of query-related entities, which is the same as the *Occurrence Ranking*. Chunks are ranked by these metrics in sequence, first by entity coverage, then by entity occurrence, and the top-$k$ are selected as supplementary evidence. This operation can be facilitated by the chunk-to-entity index $I_{c \to e}(\cdot)$ to minimize the time cost.

After retrieving all relevant chunks, we proceed to rank and format the chunks and entities as supplementary input to the LLM. Following the earlier intuition that entities serve to highlight the key information while chunks provide the supporting details, we organize the retrieved evidence in an "entity1-entity2: chunks" format. To further reduce token consumption, we apply two optimization strategies. First, to eliminate redundant input caused by chunks associated with multiple entity pairs, we consolidate these chunks into a single format such as "entity1-entity2-····-entity$n$: chunks". This de-duplication step ensures that each chunk is included only once, even if it is linked to multiple entity pairs. Second, we detect and merge continuous chunks within the evidence set to eliminate overlaps between adjacent chunks. This chunk merging step further reduces input redundancy and helps minimize token costs. Finally, we rank the entity pairs based on entity coverage and arrange their corresponding chunks according to their original chunk order in the document.

## 4 EXPERIMENT

### 4.1 EXPERIMENT SETTINGS

We describe our experimental setup, including the choice of base models, datasets, and evaluation metrics. For each component, we detail both the selection criteria and the rationale behind them, aiming to ensure the reproducibility, practicality, and fairness of our evaluation.

**Base Models** We adopt two open-source lightweight models, Qwen2.5-7B-Instruct (Qwen et al., 2025) and Llama3.1-8B-Instruct (Grattafiori et al., 2024), to ensure practicality and reproducibility under limited resources and privacy constraints. For embeddings, we use BGE-M3 (Chen et al., 2024a), a state-of-the-art open-source model. Entity extraction is exemplified by the use of SpaCy, including the ablation study, while comparisons with other extractors are reported in Appendix D.1.

**Datasets** We evaluate on QA datasets built from extremely long documents, including NovelQA (Wang et al., 2025a) and two subsets of Infinite-Bench (Zhang et al., 2024), namely InfiniteChoice and InfiniteQA. Each document averages about 200k tokens, allowing us to assess global query performance over long contexts (see Appendix B for details). We exclude UltraDomain (Qian et al., 2025) used in LightRAG due to its reliance on LLM-as-judge evaluation (Szymanski et al., 2025; Tian et al., 2023; Ye et al., 2025), which raises concerns about reliability. Instead, following RAPTOR (Sarthi et al., 2024), we focus on closed-ended QA and multiple-choice tasks for more accurate and interpretable evaluation.

**Metrics** For multiple-choice and closed-ended QA tasks, we employ accuracy and ROUGE-L (Lin, 2004), respectively. To evaluate efficiency, we measure the average indexing time for each document and the average retrieval latency per query.

Table 1: Overall results of the three datasets, the **best results** are in bold and the runner-up is underlined. Met. denotes the evaluation metric (accuracy for NovelQA/InfiniteChoice, Rouge-L for InfiniteQA). IT means indexing time per document, and QT means querying time per question.

| Backbone Model | | Qwen2.5-7B-Instruct | | | Llama3.1-8B-Instruct | | |
|---|---|---|---|---|---|---|---|
| Dateset | | NovelQA | InfiniteChoice | InfiniteQA | NovelQA | InfiniteChoice | InfiniteQA |
| GraphRAG-L | Met. ↑ | 43.34 | 46.72 | 13.51 | **43.64** | **43.66** | 6.37 |
| | IT ↓ | 13793.89 | 11816.15 | 15686.53 | 4517.09 | 3921.95 | 5533.68 |
| | QT ↓ | 0.20 | 0.25 | 0.82 | 0.43 | 0.41 | 1.16 |
| GraphRAG-G | Met. ↑ | 17.48 | 18.78 | 2.32 | 10.93 | 9.17 | 1.98 |
| | IT ↓ | 13793.89 | 11816.15 | 15686.53 | 4517.09 | 3921.95 | 5533.68 |
| | QT ↓ | 15.72 | 16.65 | 2.83 | 3.25 | 3.86 | 3.33 |
| HippoRAG2 | Met. ↑ | 44.60 | **48.91** | 8.43 | 26.03 | 20.69 | 3.58 |
| | IT ↓ | 11102.11 | 10624.65 | 13525.97 | 14279.35 | 17131.67 | 18384.76 |
| | QT ↓ | 3.81 | 2.40 | 3.10 | 2.82 | 3.84 | 2.06 |
| LightRAG | Met. ↑ | 38.57 | 45.41 | 10.41 | 21.82 | 20.52 | 3.44 |
| | IT ↓ | 5290.93 | 4732.98 | 6976.55 | 3416.31 | 3225.94 | 5231.11 |
| | QT ↓ | 15.68 | 16.03 | 15.97 | 11.44 | 12.92 | 15.44 |
| RAPTOR | Met. ↑ | 37.27 | 34.93 | 6.42 | 40.48 | 37.12 | 5.83 |
| | IT ↓ | 2847.25 | 2568.26 | 3407.41 | 2874.65 | 2551.89 | 2844.55 |
| | QT ↓ | **0.02** | 0.08 | **0.03** | **0.02** | **0.03** | **0.03** |
| E²GraphRAG | Met. ↑ | **45.60** | 43.23 | **13.65** | 41.26 | 39.74 | **11.07** |
| | IT ↓ | **1397.11** | **1244.56** | **1630.87** | **1641.49** | **1433.74** | **1839.26** |
| | QT ↓ | **0.02** | **0.05** | **0.03** | 0.03 | 0.05 | **0.03** |

## 4.2 BASELINES

We compare against all publicly available open-source methods to ensure a comprehensive evaluation, including GraphRAG-Local, GraphRAG-Global, LightRAG-Hybrid, HippoRAG2, and RAPTOR. For RAPTOR, we aligned its prompting format with ours, while for LightRAG, HippoRAG2, and GraphRAG, we adopted their default prompts with retries to address JSON extraction failures in smaller models. Further implementation details are provided in Appendix C.

## 4.3 EXPERIMENTAL RESULTS

As shown in Table 1, E²GraphRAG achieves the highest efficiency in the indexing stage, being **up to 10× faster than GraphRAG** and about **2× faster than RAPTOR**. In retrieval, E²GraphRAG also shows superior speed, reaching **over 100× faster than LightRAG** and nearly **10× faster than GraphRAG (local)**. Meanwhile, E²GraphRAG maintains effectiveness on par with GraphRAG, achieving the best performance on NovelQA with Qwen and on InfiniteQA across both backbones.

In contrast, existing baselines reveal an inherent trade-off between effectiveness and efficiency. GraphRAG achieves the highest QA accuracy, but suffers from extremely low efficiency. LightRAG and HippoRAG2 improve efficiency by simplifying the workflow and reducing some overhead, yet this comes at the cost of decreased QA accuracy. Among the baselines, RAPTOR is the most efficient, but its effectiveness is among the lowest. It's also worth noting that HippoRAG2 and LightRAG exhibit a substantial decline in performance when the base model switches from Qwen to Llama. We attribute this degradation to the variation in entity extraction capabilities across different LLMs.

In summary, existing methods involve trade-offs and lack Pareto-frontier improvements. As illustrated in Figure 3, our method (red star) consistently operates in the desired top-left quadrant, surpassing the established Pareto front in most scenarios and achieving superior cost-performance efficiency. To facilitate a clearer understanding of how our method works, we provide a case study in Appendix D.5.

## 4.4 COMPUTATIONAL COST ANALYSIS

In addition to the wall-time comparison reported in Table 1, Table 2 summarizes the average indexing cost and token usage per book on NovelQA, InfiniteChoice, and InfiniteQA, calculated using the

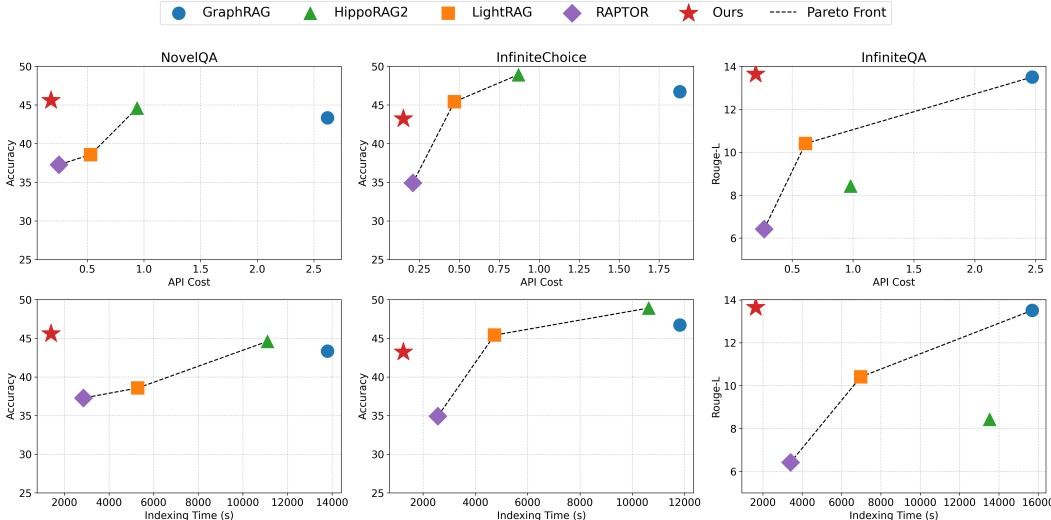

Figure 3: Comparative analysis of our method against baselines based on Qwen across three datasets. Each plot evaluates the trade-off between performance and cost. The black dashed line represents the Pareto front established by the baseline methods.

Table 2: Average indexing cost **per book** and token usage on NovelQA, InfiniteChoice, and InfiniteQA. Costs are computed using the official pricing of the Qwen2.5-7B-Instruct API. For each method, we report absolute values along with relative multiples ($\times$) compared to E$^2$GraphRAG.

| Backbone Model | | Qwen2.5-7B-Instruct | | |
|---|---|---|---|---|
| Dateset | | NovelQA | InfiniteChoice | InfiniteQA |
| GraphRAG | Input tokens ↓ | 1,684,445 ($\times$6.54) | 1,069,421 ($\times$4.70) | 1,396,578 ($\times$4.80) |
| | Output tokens ↓ | 1,778,697 ($\times$37.55) | 1,343,773 ($\times$32.58) | 1,768,985 ($\times$33.50) |
| | Cost ↓ | 2.62 ($\times$14.56) | 1.88 ($\times$12.53) | 2.47 ($\times$12.35) |
| HippoRAG2 | Input tokens ↓ | 1,236,157 ($\times$4.80) | 1,133,410 ($\times$4.98) | 1,237,688 ($\times$4.25) |
| | Output tokens ↓ | 319,988 ($\times$6.75) | 297,295 ($\times$7.21) | 358,637 ($\times$6.79) |
| | Cost ↓ | 0.94 ($\times$5.22) | 0.87 ($\times$5.87) | 0.98 ($\times$4.9) |
| LightRAG | Input tokens ↓ | 713,782 ($\times$2.77) | 621,740 ($\times$2.73) | 795,834 ($\times$2.74) |
| | Output tokens ↓ | 173,186 ($\times$3.66) | 156,146 ($\times$3.78) | 212,674 ($\times$4.03) |
| | Cost ↓ | 0.53 ($\times$2.94) | 0.47 ($\times$3.13) | 0.61 ($\times$3.05) |
| RAPTOR | Input tokens ↓ | 323,233 ($\times$1.26) | 277,796 ($\times$1.22) | 362,129 ($\times$1.24) |
| | Output tokens ↓ | 88,889 ($\times$1.88) | 71,582 ($\times$1.74) | 92,138 ($\times$1.74) |
| | Cost ↓ | 0.25 ($\times$1.39) | 0.21 ($\times$1.40) | 0.27 ($\times$1.35) |
| E$^2$GraphRAG | Input tokens ↓ | 257,500 | 227,474 | 290,923 |
| | Output tokens ↓ | 47,375 | 41,244 | 52,800 |
| | Cost ↓ | 0.18 | 0.15 | 0.20 |

official pricing of the Qwen2.5-7B-Instruct API. For clarity, we report both the absolute values and relative multiples compared to E$^2$GraphRAG, which provides further evidence of the superior efficiency of our approach. As a complement, we provide a more intuitive visualization of indexing efficiency in Appendix D.4, which presents scatter plots of indexing time across varying document lengths based on the Qwen model. Each method is fitted with a linear function to highlight the differences in time overhead, indicating that our method scales linearly with the lowest slope among all methods.

Furthermore, to better understand the computational burden, we estimate the theoretical costs associated with these results. As the primary expense in both indexing and querying arises from LLM inference, we derive the number of LLM calls required by each method and report them in the Table 4. This result shows that E$^2$GraphRAG significantly lowers the cost in both stages. Details of the theoretical estimation are provided in Appendix D.2 and Appendix D.3.

Table 3: Ablation study results. The best results for each dataset are highlighted in bold. For other methods, the performance difference compared to E$^2$GraphRAGis annotated below each value, with ↓ (in red) indicating a decrease and ↑ (in green) indicating an increase. The annotated numbers represent the absolute difference in performance relative to E$^2$GraphRAG.

| Dataset Metric | NovelQA Acc. | InfiniteChoice Acc. | InfiniteQA R-L |
|---|---|---|---|
| E$^2$GraphRAG | 45.38 | **43.23** | **13.65** |
| Dense Retrieval Only | 42.00 (↓ 3.38) | 41.04 (↓ 2.19) | 10.03 (↓ 3.62) |
| w/o Graph Filter | 44.30 (↓ 1.08) | 36.68 (↓ 6.55) | 10.47 (↓ 3.18) |
| w/o Entity-Aware Ranking | 44.12 (↓ 1.26) | 40.17 (↓ 3.06) | 8.25 (↓ 5.40) |
| w/o Graph Filter & Entity-Aware Ranking | 44.08 (↓ 1.30) | 35.81 (↓ 5.23) | 10.58 (↓ 3.07) |
| w/o Dense Retrieval | **45.90** (↑ 0.52) | 37.99 (↓ 5.24) | 13.03 (↓ 0.62) |
| w/o Occurrence Ranking | 44.25 (↓ 1.13) | 37.99 (↓ 5.24) | 8.39 (↓ 5.16) |
| w/o Dense Retrieval & Occurrence Ranking | 45.33 (↓ 0.05) | 37.55 (↓ 5.68) | 11.07 (↓ 2.58) |

## 4.5 ABLATION STUDY

To thoroughly evaluate the contribution of each component in E$^2$GraphRAG, we conduct a comprehensive ablation study on three datasets using the Qwen model. The results are summarized in Table 3, which consists of three main sections:

**Baseline Dense Retrieval Only:** To verify the necessity and effectiveness of our retrieval strategy, we compare E$^2$GraphRAG with a baseline that relies solely on dense retrieval and the built summary tree. The results demonstrate that E$^2$GraphRAG significantly outperforms this baseline, validating the importance of our retrieval enhancements.

Table 4: Comparison of theoretical LLM calls across methods, where $n$ is the number of chunks and $m$ is the community counts for GraphRAG. $C_{\text{window}}$ is the length of LLMs' context window.

| Method | Indexing | Query |
|---|---|---|
| GraphRAG | $n + m$ | $m \times \text{len}(m)/C_{\text{window}}$ |
| LightRAG | $n$ | 1 |
| HippoRAG2 | $2n$ | 1 |
| RAPTOR | $\geq \lceil n/(g-1) \rceil$ | 0 |
| **E$^2$GraphRAG** | $\lceil n/(g-1) \rceil$ | 0 |

**Local Retrieval Ablations:** To assess the impact of the local retrieval components, we individually and jointly ablate the *Graph Filter* and *Entity-Aware Ranking* modules. Results show that both modules are crucial for local evidence selection. The removal of either leads to a significant performance drop, confirming their complementary roles.

**Global Retrieval Ablations:** Similarly, we evaluate the contribution of the global retrieval by ablating *Dense Retrieval* and *Occurrence Ranking*. Among these, Occurrence Ranking appears more impactful, likely due to its more frequent use in our datasets. Interestingly, we observe an anomalous improvement when removing Dense Retrieval on NovelQA. We hypothesize that this is caused by occasional hallucinations, where the model guesses the correct answer without actual evidence.

## 5 CONCLUSION

In this paper, we addressed the inefficiency of existing graph-based RAG methods that hinders their practicality. We streamlined the graph-based RAG pipeline and propose E$^2$GraphRAG. During the indexing stage, we recursively built document summary trees with LLMs and efficiently extracted entity-level knowledge graphs using traditional NLP toolkits such as SpaCy, BERT, and NLTK, significantly reducing time costs and improving practicality. In the retrieval stage, we proposed an adaptive strategy that leverages the graph structure to locate relevant chunks and automatically select between local and global retrieval modes, eliminating the need for manually pre-defined query settings. By combining the summary tree and knowledge graph, E$^2$GraphRAG enables adaptive global and local retrieval. Extensive experiments demonstrate that E$^2$GraphRAG achieves state-of-the-art efficiency in both indexing and retrieval stages, with up to 10× speedup over GraphRAG in indexing and 100× over LightRAG in retrieval, while maintaining comparable effectiveness.

REPRODUCIBILITY

Our code is available at `https://anonymous.4open.science/r/E-2GraphRAG-8897` with the configuration files for reproducing our results. We also provide the pseudo-code in Appendix A.

ETHICS STATEMENT

Our work proposes a more efficient and effective graph-based retrieval-augmented generation (RAG) framework, which may benefit downstream applications such as open-domain question answering, knowledge-intensive NLP tasks, and long-document understanding. By significantly reducing the indexing and retrieval cost, our approach could improve the accessibility of large-scale knowledge systems in low-resource or cost-sensitive settings.

However, like other RAG-based systems, our model depends heavily on the quality and neutrality of the underlying documents. If biased or incorrect data are indexed, the system may generate misleading or harmful outputs.

While we do not directly address issues such as fairness or bias mitigation, we encourage responsible use of our framework in conjunction with trustworthy data sources and human oversight. Future work could explore debiasing methods and improved transparency in retrieval paths.

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

## A PSEUDO CODE

---

**Algorithm 1** The pseudo-code for retrieval stage.

---

**Require:** $q, \mathcal{G}, T, k, h, l$
**Ensure:** Supplemental text $\mathcal{C}_s = \{c_1, c_2 \cdots c_n\}, n \leq k$
1: entities $\mathcal{E}_q = \texttt{SpaCy}(q)$
2: **if** $\text{Count}(\mathcal{E}_q) == 0$ **then**
3:     **return** ★ $\mathcal{C}_s = \texttt{DenseRetrieval}(q, k)$
4: **end if**
5: selected pairs $\mathcal{P}_h = \texttt{GraphFilter}(\mathcal{E}_q, h)$    [Equation 1]
6: **if** $\text{Count}(\mathcal{P}_h) == 0$ **then**
7:     ★ candidate supplementary chunks $\hat{\mathcal{C}} = \texttt{DenseRetrieval}(q, 2k)$
8:     **return** ★ $\mathcal{C}_s = \texttt{OccurrenceRank}(\hat{\mathcal{C}})$
9: **end if**
10: candidate supplementary chunks $\hat{\mathcal{C}} = \texttt{IndexMapping}(\mathcal{P}_h)$    [Equation 2]
11: **while** $\text{Count}(\hat{\mathcal{C}}) > 25$ **do**
12:     $h = h - 1$ or $l = l + 1$
13:     $\hat{\mathcal{C}}_{\text{prev}} = \hat{\mathcal{C}}$
14:     $\mathcal{P}_h = \texttt{GraphFilter}(\mathcal{E}_q, h)$    [Equation 1]
15:     $\hat{\mathcal{C}} = \texttt{IndexMapping}(\mathcal{P}_h)$    [Equation 2]
16: **end while**
17: **if** $\text{Count}(\hat{\mathcal{C}}==0)$ **then**
18:     **return** $\mathcal{C}_s = \texttt{EntityAwareFilter}(\hat{\mathcal{C}}_{\text{prev}})$
19: **else**
20:     **return** $\mathcal{C}_s = \hat{\mathcal{C}}$
21: **end if**

---

## B DATASET DETAILS

In this section, we provide detailed descriptions of the datasets used in our experiments. While the main paper introduces the overall dataset choices and their relevance to our task, here we include further information on data statistics.

**NovelQA** has 89 books along with 2305 multiple-choice questions in total, which contain 65 free public-domain books and 24 copyright-protected books purchased from the Internet. It is released with an Apache-2.0 License. **InfiniteChoice** has 58 books along with 229 multiple-choice questions in total. **InfiniteQA** has 20 books along with 102 questions in total. The InfiniteBench is released with an MIT License. The links to the two datasets are provided in our code repository, and both datasets are publicly accessible.

To provide a deeper insight into our method, we analyze the number of entities involved in each question for all datasets. Specifically, we count the entities mentioned in the question text, excluding those appearing only in the multiple-choice options. The detailed statistics, including the average, minimum, and maximum number of entities per question, are reported in Table 5. In addition, Figure 4 illustrates the distribution of questions across different entity count buckets, offering a clearer view of how entity complexity varies across the datasets.

Table 5: Entity count of each question in each dataset with SpaCy as the extractor.

| Dataset | NovelQA | InfiniteChoice | InfiniteQA |
|---------|---------|----------------|------------|
| avg. | 4.60 | 3.24 | 3.37 |
| min. | 0 | 1 | 0 |
| max. | 24 | 9 | 7 |

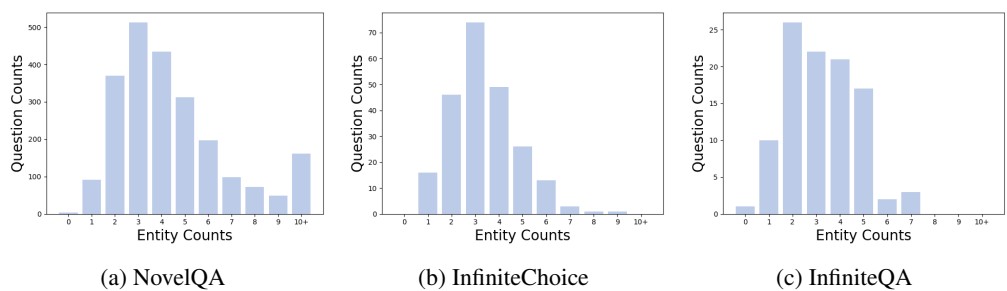

(a) NovelQA          (b) InfiniteChoice          (c) InfiniteQA

Figure 4: Distribution of questions across different entity count buckets with SpaCy as the extractor.

## C  IMPLEMENTATION DETAILS OF BASELINE METHODS

Because of excessive redundant design in the official GraphRAG implementation, we opted for the most widely adopted open-source implementation, nano-GraphRAG [2] for our experiments. To adapt GraphRAG for local deployment with Huggingface models, we utilized the code from LightRAG that supports Huggingface integration and embedded it into nano-GraphRAG.

For a fair comparison, the hyperparameter settings of all the methods and all baselines are chosen to ensure that the entire pipeline can run on a single NVIDIA A800 GPU with 80GB of memory. For the retrieval level of GraphRAG, we choose the best level, i.e., level 2, reported in the corresponding paper. For the retrieval mode of LightRAG, we choose the Hybrid mode, which is reported as the best mode in the paper. All LLMs are implemented using the HuggingFace `transformers` framework, with `temperature=0.7` and `max_new_tokens=1200` for summarization or entity extraction, except for the GraphRAG, which requires raising the `max_new_tokens` to 8192 for generating a complete JSON structure. For our method, the configuration files for reproducing the results have been provided on GitHub, with the hyperparameter $k = 25$, which is determined by the GPU memory, and $h$ is automatically adjusted during retrieval; therefore, we choose a relatively large value of 4.

## D  SUPPLEMENTARY ANALYSIS

### D.1  COMPARISON BETWEEN DIFFERENT EXTRACTORS

Table 6: Overall results with different entity extractors on NovelQA, InfiniteChoice, and InfiniteQA. Met. means the metric for each dataset, and ET means extracting time.

| Backbone Model | | Qwen2.5-7B-Instruct | | | Llama3.1-8B-Instruct | | |
|---|---|---|---|---|---|---|---|
| Extractor | | NovelQA | InfiniteChoice | InfiniteQA | NovelQA | InfiniteChoice | InfiniteQA |
| NLTK | Met. ↑ | 46.77 | 40.61 | 16.99 | 40.13 | 37.12 | 10.18 |
| | ET ↓ | 2577.14 | 2315.52 | 3286.00 | 2511.18 | 2241.62 | 3263.51 |
| BERT | Met. ↑ | 45.94 | 37.99 | 12.60 | 39.92 | 37.99 | 13.21 |
| | ET ↓ | 58.76 | 60.42 | 84.64 | 63.19 | 60.48 | 84.24 |
| SpaCy | Met. ↑ | 45.60 | 43.23 | 13.65 | 41.26 | 39.74 | 11.07 |
| | ET ↓ | 39.82 | 35.15 | 45.26 | 38.47 | 33.26 | 46.43 |

To save computing resources and ensure a fair comparison, we do not rebuild the summary tree for different extractors. Instead, we apply each extractor to the same summary tree and report the extraction time and performance in the Table 6. The performance of different extractors is relatively close, and all remain competitive. With the support of modern GPUs, the BERT-based extractor also demonstrates high efficiency. In contrast, the NLTK extractor exhibits low efficiency, since it is primarily a non-industrial toolkit. The SpaCy extractor achieves the highest efficiency while maintaining independence from GPU resources.

---

[2] https://github.com/gusye1234/nano-graphrag

## D.2 COMPARISON ON INDEXING COST OF DIFFERENT METHODS

Following the symbols defined in Table 4, we use $n$ to denote the number of chunks, $m$ for the number of communities detected by GraphRAG, and $g$ represents the group size (or maximum group size) for E$^2$GraphRAG and RAPTOR.

**E$^2$GraphRAG** Our method builds a tree with $n$ leaf nodes and each non-leaf node has $g$ child nodes. The number of LLM calls is equal to the number of non-leaf nodes. The non-leaf nodes can be listed by level and form a geometric sequence with the first term $\lceil \frac{n}{g} \rceil$ and the common ratio $\frac{1}{g}$, where each term is rounded up to the nearest integer. Consequently, the total number of LLM calls can be expressed as:

$$S = \sum_{k=1}^{K} \left\lceil \frac{n}{g^k} \right\rceil,$$

where $K = \lceil \log_g n \rceil$. Since $\lceil x \rceil \in (x, x+1]$, the upper and lower bounds of the equation above can be derived as:

$$\sum_{k=1}^{K} \frac{n}{g^k} < S \le \sum_{k=1}^{K} \frac{n}{g^k} + K$$

Applying the geometric series sum formula $S = a\frac{1-r^n}{1-r}$, the equation above can be rewritten as:

$$\frac{n}{g-1}\left(1 - g^{-K}\right) < S \le \frac{n}{g-1}\left(1 - g^{-K}\right) + K$$

Because $K$ grows only logarithmically with respect to $n$, its contribution to the overall number of LLM calls is negligible. For simplicity, we therefore approximate the count by $n/(g-1)$.

**RAPTOR** Similar to our method, RAPTOR builds a tree with $n$ leaf nodes. However, each non-leaf node has at most $g$ child nodes, resulting in the number of non-leaf nodes being larger than E$^2$GraphRAG. Therefore, the **lower bound** is $n/(g-1)$.

**LightRAG** LightRAG extracts the entities and relations from each chunk and then assembles them into an entity graph. The extraction phrase takes $n$ times LLM calls, and the assembling phrase does not need the LLM calls. Therefore, it requires $n$ times of LLM calls in total.

**GraphRAG** GraphRAG extracts the entities and relations from each chunk and formats an entity graph. Then, GraphRAG clusters the nodes and summarizes each community to aggregate the information by LLM. Therefore, the extraction phrase evokes $n$ times LLM calls, and the summarization phrase calls LLM $m$ times, which is $n + m$ in total.

**HippoRAG2** HippoRAG2 extracts the entities first and then extracts relations from each chunk. It takes $2n$ times LLM calls. It's worth noting that HippoRAG2 does not generate the tedious entity descriptions or relation descriptions, which significantly reduces the token cost compared to LightRAG and GraphRAG.

## D.3 COMPARISON ON QUERYING COST OF DIFFERENT METHODS

Following the symbols defined in Section 3 and Table 4, we use $n$ to denote the number of chunks, $m$ for the number of communities detected by GraphRAG, $g$ for the group maximum group size, and $k$ to represent the maximum number of chunks to retrieve.

**E$^2$GraphRAG** In the query stage, the primary computational overhead arises from entity extraction, whose cost varies with the efficiency of the extractor employed. In comparison, the graph search introduces only minor latency, which remains negligible relative to the time consumed by LLM invocations.

**RAPTOR** The primary computational overhead of RAPTOR lies in the dense embedding retrieval, which relies on GPUs for acceleration.

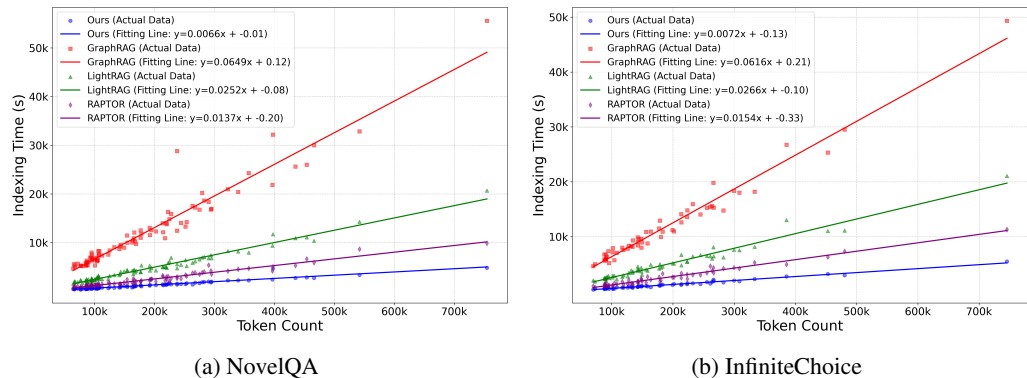

(a) NovelQA  (b) InfiniteChoice

Figure 5: Time cost as a function of document token count for each method. The statistic is based on NovelQA and InfiniteChoice with Qwen as the base model.

**LightRAG**  LightRAG uses the LLM to extract entities from the query; hence, the overhead is dominated by the LLM invocation.

**GraphRAG**  In local mode, GraphRAG performs dense retrieval to identify the most similar nodes. In global mode, it relies on the LLM to select relevant communities for answering the query, resulting in $m \times \text{len}(m)/C_{\text{window}}$ LLM calls.

**HippoRAG2**  HippoRAG2 employs the LLM for reranking, so the query-time cost is primarily driven by LLM calls.

### D.4 VISUALIZATION OF INDEXING EFFICIENCY

To provide a more intuitive comparison of how indexing time scales with the size of the text corpus, we fitted a function to the indexing time versus text length data for all four methods, as shown in Figure 5. The $R^2$ values for each method on every dataset exceed 0.90, indicating a strong goodness-of-fit. From the figure, it is evident that all four methods exhibit approximately linear growth, while our method demonstrates **the lowest slope**. This observation aligns with both our theoretical analysis and experimental results, providing clear evidence that our approach **scales linearly with the knowledge base size at the minimal rate among the compared methods.**

### D.5 CASE STUDY

To better illustrate how our method operates, we present two case studies under different query modes, as shown in Figure 6. In both cases, named entities are first extracted and then mapped onto the graph constructed during the indexing stage. For the case on the left, among the five extracted entities, three are absent from the graph, while two are connected. We identify the shortest path, *Quirrell - Harry - Change*, which enables us to localize the relevant chunks based on these entities. This leads us to chunks simultaneously associated with *Quirrell* and Change, from which we uncover key evidence that *Quirrell* attempted to kill Harry during the Quidditch match.

For the case on the right, only one of the three entities appears in the graph. Hence, we employ the global query mode, performing dense retrieval over the entire summary tree and ranking candidates by the frequency of the entity Voldemort. We observe that $c_4$ appears 4 times, $c_6$ appears 2 times, and $c_8$ appears 3 times, while $s_2$ accumulates $c_4 + c_5 + c_6 = 4 + 0 + 2 = 6$ occurrences. We ultimately select $c_4$ and $s_2$, and within the summary of $s_2$, we identify an alias of Voldemort generated by the LLM.

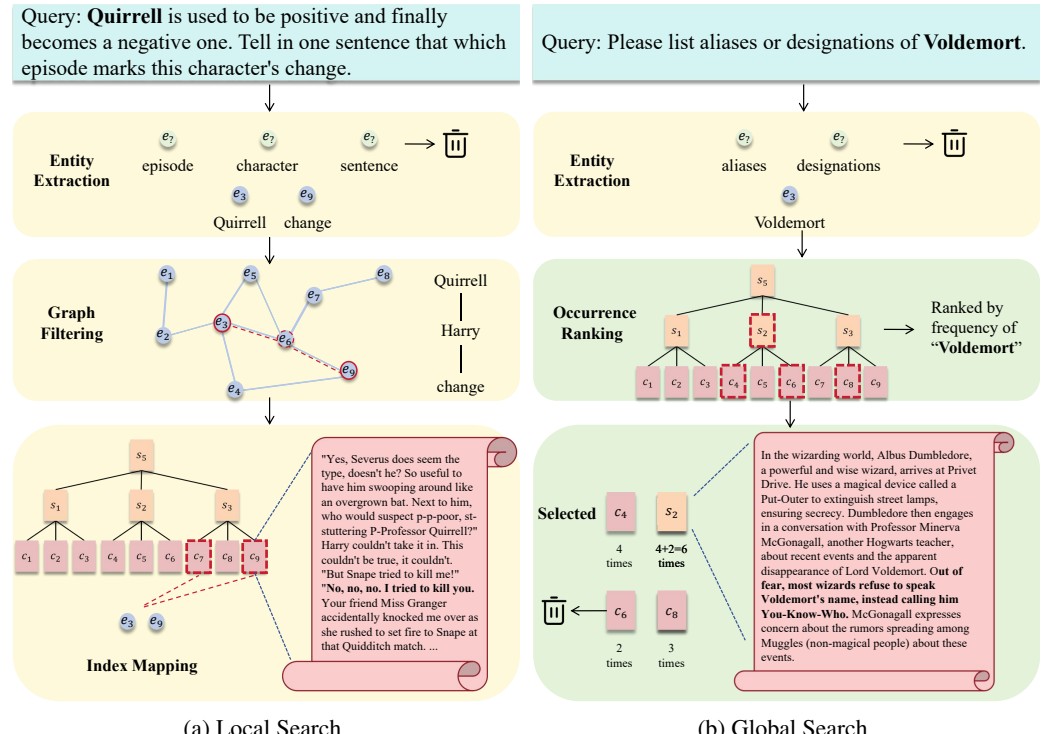

Figure 6: Case study examples: local search (left) vs. global search (right).

## E    LIMITATIONS

Although we present a streamlined graph-based RAG framework that demonstrates both strong efficiency and effectiveness in this paper, the retrieval design remains relatively intuitive. While we have conducted extensive experiments and explored various alternative retrieval strategies (some of which are not included in the paper), it is impossible to exhaust all possible retrieval pipeline designs. Therefore, there may still exist more optimal retrieval strategies that could further enhance the performance of our approach.

## F    THE USE OF LLMS

This paper employed LLMs solely for grammatical correction and stylistic refinement, with the purpose of more effectively communicating our results and conclusions.

