# OpenReview forum: "E$^2$GraphRAG: Advancing the Pareto Frontier in Efficiency and Effectiveness for Graph-based RAG"
_ICLR.cc/2026/Conference — Submitted to ICLR 2026_

### Official Review · Reviewer_Zxuz · 2025-10-28

**Soundness:** 3
**Presentation:** 3
**Contribution:** 2
**Rating:** 4
**Confidence:** 5

**Summary:**

$E^2GraphRAG$ is a streamlined framework for graph-based Retrieval-Augmented Generation that aims to advance the Pareto frontier of efficiency and effectiveness of the GraphRAG paradigm. The approach enables efficient graph construction and retrieval while maintaining superior performance. Constributions inlcude:
1. integration of summary trees and entity graphs for lightweight indexing
2. adaptive retrieval strategy that automatically selects query modes using graph structure

**Strengths:**

1. Efficiency Improvements. By replacing LLM-based entity extraction with standard NLP tools and simplifying extracting specific relation to sentence-level "co-occurrence", the method reduces indexing time and computational overhead, making it more practical for large-scale knowledge base.
2. Adaptive Retrieval. The graph-filtering approach for mode selection is innovative and flexible.
3. Strong Empirical Validation, as shown by the experiments.

**Weaknesses:**

While the proposed model appears novel and promising, I identified several contradictions and points of confusion that could undermine the paper's overall quality and the replicability of its conclusions.

1.  Usage of SpaCy Sacrifices Generalizability Without Reducing Overall Complexity

In discussing the motivation for adopting SpaCy-based NER, the authors claim that it reduces computational costs compared to LLM-based alternatives. However, the construction of the summary tree still necessitates processing the entire corpus through LLMs for recursive summarization, rendering the LLM token costs largely unavoidable. Furthermore, the complexity of LLM-based NER would be approximately $O(N \cdot T)$ (where $N$ is the number of chunks and $T$ is the average token size per chunk). This is comparable to the costs incurred by the summary tree construction. It is not the primary bottleneck relative to those costs. Consequently, employing SpaCy does not alter the overall token complexity of the indexing stage.

It is also worth noting that SpaCy-based NER can introduce noise and often struggles with domain-specific corpora, potentially compromising the accuracy of entity extraction in specialized contexts (e.g. legal, finance, medical).

2. Limited novelty in indexing stage

The summary tree construction is similar to the community summarization in GraphRAG[1] and the recursive summarization in Raptor[2], which has been widely used.

3. When building the summary tree, the chunks are grouped by taking the consecutive $g$ chunks following the order in the original passages. How can this be generalize to the domain where this bias is not available? For instance, in a knowledge base where all chunks are independent passages. If you use clustering to group chunks in this scenario, the efficiency may be an issue.

- [1] From Local to Global: A Graph RAG Approach to Query-Focused Summarization, arxiv-2404.16130
- [2] RAPTOR: Recursive Abstractive Processing for Tree-Organized Retrieval, ICLR, 2024

**Questions:**

Plz see above.

---

> ### Author Response · Authors · 2025-11-14
>
> Thank you for your suggestions. Here is our response.
>
> **W1. Usage of SpaCy Sacrifices Generalizability Without Reducing Overall Complexity**
>
> Thank you for your question, but we believe there exists **factual error** in your analysis.
>
> First, we have provided the time cost and token usage in Tables 1 and 2, respectively. These results show that our method requires fewer tokens than all baselines, demonstrating its efficiency.
>
> Second, we also provide theoretical estimations in Appendices D.2 and D.3.
>
> Third, the complexity calculation in your comment ignores two important factors: the number of input and output tokens. For LLMs, if the prompt length is denoted as $n$, the inference complexity for **each** new token is $O(n^2)$.
>
> - **Input tokens**: When extracting entities with LLMs, a long few-shot prompt is typically required to ensure high-quality output. In contrast, summarization does not require such long prompts, as noted in Lines 159–161 of our paper.
>
> - **Output tokens**: Entity extraction requires the LLM to produce JSON-formatted outputs to facilitate subsequent processing, which includes substantial extra tokens for maintaining the JSON structure. Summarization, by contrast, only requires generating a string, significantly reducing extra output tokens. By empirical observation and reports from LightRAG’s GitHub issues, the token generation limit sometimes required to be set to 8192 to produce valid JSON for LightRAG, whereas our method generates fewer than 1200 new tokens.
>
> In summary, our method inputs fewer tokens, outputs fewer tokens, and makes fewer LLM calls, which explains the observed superior efficiency.
>
> **W2. Limited novelty in indexing stage**
>
> We kindly refer you to the global response for our explanation on novelty.
>
> **W3. Method Generalization**
>
> Thank you for your suggestion. To evaluate generalization, we conducted experiments on HotpotQA (randomly sampling 1,000 entries from the validation set), where each chunk corresponds to a single document (wiki page), representing an open-domain scenario. For domain-specific evaluation, we performed experiments on the medical dataset, a subset of GraphRAG-Bench [1]. All baseline methods share the same backbone model, Llama3-8B, to ensure a fair comparison. For HotpotQA, we report Contain-Match (HotpotQA-Contain) and GPT-Evaluation (HotpotQA-GPT) Accuracy, while for the medical dataset, we use GPT-Evaluation Accuracy (Medical-GPT) only, as it presents greater challenges for automatic evaluation.
>
> The experimental results are summarized below:
>
> |      | HotpotQA-Contain | HotpotQA-GPT | Medical-GPT |
> |------|------|------|------|
> | RAPTOR | 55.90  | 58.30 | 55.75 |
> | LightRAG | 60.30  | 59.50 | 54.36 |
> | HippoRAG | 57.00 | 59.30 | 55.04 |
> | E$^2$GraphRAG | **61.00** | **63.90** | **58.00** |
>
> The results show that, although our method does not include any specific design for multi-hop questions, it still achieves **top-tier performance**. We attribute this to the **inherent multi-hop reasoning capability** introduced by the entity graph structure.
>
> For the **domain-specific questions**, our method also outperforms all baselines. This improvement likely stems from the fact that large language models tend to **hallucinate when encountering domain-specific entities**, whereas our framework helps mitigate this issue. Specifically, our framework assumes the entity graph is inherently noisy from the very beginning. Therefore, we use it merely as an entity filter, which reduces reliance on the graph’s accuracy and consequently leads to better overall performance.
>
> Moreover, adapting LLMs to perform well on domain-specific named entity recognition (NER) is often challenging. In contrast, our approach allows easy adaptation by tuning a BERT-based NER model or leveraging off-the-shelf domain-specific NER systems, making it both practical and efficient.
>
>
> [1] Xiao et al., GraphRAG-Bench: Challenging Domain-Specific Reasoning for Evaluating Graph Retrieval-Augmented Generation

---

> ### Comment · Reviewer_Zxuz · 2025-11-27
> **In response to Author (Submission17284)'s rebuttal.**
>
> Dear Authors,
>
> Thank you for the detailed response. However, some of my concerns remain unaddressed:
>
> 1. Regarding the complexity of your algorithm, I previously noted that your use of SpaCy for NER does not reduce the overall big O complexity of token cost, as the summary tree still dominates the computational expense. The fundamental big O complexity remains unchanged, which is also supported by your comparison with Raptor in Table 2.
>
> 2. Why does Raptor demonstrate higher token consumption compared to $E^2{graphRAG}$? Given that the core token consumption components should be similar, does this difference stem from engineering optimizations, prompt simplification, or other factors?
>
> 3. I did not find a "Medical" dataset referenced in the GraphRAG-Bench[1] paper you cited. Please verify this information and correct me if I'm mistaken.
>
> **References**
>
> [1] Xiao et al., GraphRAG-Bench: Challenging Domain-Specific Reasoning for Evaluating Graph Retrieval-Augmented Generation

---

> > ### Author Response · Authors · 2025-11-27
> >
> > Thank you for your reply! Here is our answer.
> >
> > **C1. Complexity of Our Algorithm**
> >
> > Then, in the comparison with RAPTOR in Table 2, our method achieves lower cost; RAPTOR’s cost is approximately 1.4× that of ours. Second, because RAPTOR requires clustering and the number of chunks per cluster varies, its computational cost, under the same maximum group size, is bounded below by the complexity of our method, as analyzed in Appendix D.2. Third, Table 1 further shows that our method outperforms RAPTOR in effectiveness and time efficiency. In summary, our approach surpasses RAPTOR in both efficiency and performance.
> >
> > Regarding your question about whether using SpaCy-based NER reduces time complexity: RAPTOR itself does not build graphs, so the appropriate comparison is with graph-based systems such as LightRAG, GraphRAG, and HippoRAG. We construct co-occurrence graphs using SpaCy, while these methods rely on LLMs to extract nodes and relations. This substantially reduces the time complexity. Even after including the document summarization tree, the total computational cost is still reduced by several-fold. A detailed analysis can be found in Appendix D.2.
> >
> > **C2. Why RAPTOR Costs More**
> >
> > RAPTOR incurs higher token costs for two main reasons:
> >
> > a. Because of clustering, chunks within a RAPTOR cluster are not necessarily consecutive. In contrast, we group consecutive chunks for summarization, allowing us to merge overlapping tokens between adjacent chunks and thus reduce token usage. For example, consider 200 chunks, each with 1200 tokens, with 100-token overlaps between adjacent chunks to avoid semantic discontinuity. A group of g consecutive chunks requires only
> > $1200×g−(g-1)×100$ tokens in our method. RAPTOR, however, would additionally process the extra $100\times (g-1)$ overlapping tokens, resulting in wasted token budget.
> >
> > b. We aggregate every $g$ nodes deterministically, whereas RAPTOR uses an upper bound of $g$, which is explicitly demonstrated in line 921-922. With a fixed number of leaf nodes, this design yields a shallower tree for our method, while RAPTOR requires more LLM calls to construct a deeper hierarchy.
> >
> >
> > **C3. Wrong Citation**
> >
> > We sincerely apologize for the oversight. We mistakenly cited reference [1], while our dataset actually comes from [2], a concurrent work, rather than from [1].
> >
> > [1] Xiao et al., GraphRAG-Bench: Challenging Domain-Specific Reasoning for Evaluating Graph Retrieval-Augmented Generation
> >
> > [2] Xiang et al., When to use Graphs in RAG: A Comprehensive Analysis for Graph Retrieval-Augmented Generation

---

> ### Comment · Reviewer_Zxuz · 2025-11-27
>
> From your experimental results, under the worst-case scenario, Raptor incurs a 40% higher cost than your method.
>
> The reason I inquired about Big O complexity is to better understand the scalability of your algorithm compared to baselines, which would help assess its advantages when applied to large knowledge graphs.
>
>
> While I acknowledge that Raptor's efficiency is theoretically upper-bounded by $E^2GraphRAG$, I wonder whether this efficiency gap remains consistent in large-scale corpora. Specifically, if the cost difference remains relatively constant (e.g., maintaining around 1.4x ratio) rather than demonstrating fundamental distinctions in computational complexity (such as between sublinear and linear scaling, or quadratic versus linear growth) what would be the compelling practical advantage of choosing $E^2GraphRAG$ over a more straightforward approach like Raptor?
>
>
> Additionally, I find it puzzling that Raptor demonstrates higher query latency compared to your method. This seems counterintuitive, as Raptor primarily relies on direct dense search in the embedding space, whereas your approach involves a more sophisticated, loop-based retrieval process that would naturally be expected to yield higher efficiency.

---

> > ### Author Response · Authors · 2025-11-27
> >
> > Thank you for your response. We further clarify the following points:
> >
> > 1. For any method, all input data, namely $N\times T$ tokens—must inevitably be processed by the model, and this cost cannot be avoided. Compared with RAPTOR, our advantage lies in the fact that we can aggregate chunks using a fixed group size $g$, while also merging overlaps between adjacent contexts, thereby reducing the total token consumption. While we acknowledge that this does not produce the kind of “fundamental distinction” in asymptotic behavior such as sublinear versus linear scaling, Figure 5 in our paper shows that we effectively reduce the slope of the scaling curve. We expect this benefit to become more pronounced on larger datasets. Moreover, our method is both more cost-efficient and more effective, which we believe makes it a more practical solution.
> >
> > 2. In practice, the retrieval latency of RAPTOR is almost the same as ours. We attribute this to differences in hardware assumptions. As illustrated in the official repo of RAPTOR, `FaissRetriever.py`, line 125-126:
> > ```
> > self.index = faiss.IndexFlatIP(self.embeddings.shape[1])
> > self.index.add(self.embeddings)
> > ```
> > RAPTOR relies on `faiss`, embedding-based retrieval executed on GPUs, which makes its search extremely fast. In contrast, our method involves additional looping and comparisons that run on CPUs, leading to slightly slower operations. Since neither approach invokes an LLM during retrieval, both methods exhibit retrieval delays on the order of $10^{-2}$ seconds. We believe this difference is negligible in practice.

---

> > > ### Comment · Reviewer_Zxuz · 2025-11-27
> > >
> > > Hi Authors,
> > >
> > > I don't believe my concerns have been fully addressed:
> > >
> > > The token reduction achieved by your method compared to Raptor appears to stem from eliminating the need to re-process overlapping contexts. While this approach has merit, it relies on the assumption that each group is formed from consecutive chunks. As I noted in my initial review, in general scenarios where chunks are independent, this assumption may not hold. Furthermore, leveraging this characteristic seems more like an engineering optimization rather than a generalizable methodological advancement. I therefore maintain my reservations regarding the efficiency improvements of your approach in broader applications.
> > >
> > > Additionally, with the ongoing expansion of LLM context windows, most documents, logs, and information pieces can now be processed in their entirety by LLMs. This suggests that text chunks in knowledge bases may increasingly maintain completeness and interdependence. In such a scenario, the scaling advantages of $E^2GraphRAG$ might not be sustained.
> > >
> > > I believe there may be a misunderstanding regarding my previous point. As you correctly noted, your method would be expected to demonstrate higher query latency than Raptor. However, in Table 1 (InfiniteChoice dataset), your reported query time is actually lower than Raptor's. This observation contradicts your theoretical analysis and appears counter-intuitive from a technical perspective.

---

> > > > ### Author Response · Authors · 2025-11-27
> > > >
> > > > Thank you for your response. However, we believe there may be some misunderstanding regarding our contributions. The core contribution of our method is advancing the Pareto frontier of GraphRAG-style approaches: our method improves both efficiency and effectiveness simultaneously.
> > > >
> > > > Compared with RAPTOR, our efficiency gains are, as you noted, not of the “fundamental distinction” type such as reducing from quadratic to linear complexity. However, within this family of methods, the overall cost typically scales linearly with corpus size. In this regime, reducing the slope of the scaling curve becomes crucial. Our method achieves the lowest slope among the compared approaches, indicating stronger scalability to larger corpora. Combined with our experimental results—where our method outperforms RAPTOR across every benchmark—we believe the empirical evidence clearly supports both our efficiency improvements and our superior effectiveness.
> > > >
> > > > Next, regarding your point about chunks being independent, our method still applies in that setting. As described in points C2–b of our previous response, our approach constructs a flatter tree, reducing the number of LLM calls and lowering token consumption. Furthermore, we recorded indexing time on multi-hop QA datasets, where each chunk corresponds to an independent document, and our method still achieves better performance than RAPTOR.
> > > >
> > > > |      | Indexing Time | Retrieval Time |
> > > > |------|------|------|
> > > > | RAPTOR | 1323.57  | 0.062 |
> > > > | LightRAG | 4933.22  | 10.963 |
> > > > | HippoRAG | 936.00 | 1.461 |
> > > > | E$^2$GraphRAG | **534.60** | **0.053** |
> > > >
> > > > Finally, concerning the retrieval latency discussion, our earlier explanation may have caused some confusion. Our intention was not to claim that our method is inherently slower than RAPTOR in terms of latency. On the contrary, our theoretical computational cost and number of operations are indeed lower. However, because different operations have different execution times on real hardware—and because system noise can introduce additional variance—the empirical retrieval latencies of the two methods do not always reflect the theoretical advantage directly. As a result, the two methods exhibit comparable latency on some datasets, while each occasionally leads on others.

---

### Official Review · Reviewer_XF2c · 2025-10-29

**Soundness:** 3
**Presentation:** 3
**Contribution:** 3
**Rating:** 6
**Confidence:** 3

**Summary:**

This paper proposes E2GraphRAG, a graph-based retrieval-augmented generation (RAG) framework that improves both efficiency and effectiveness over existing methods such as GraphRAG and LightRAG. The key idea is to combine a hierarchical summary tree (built by LLM summarization) with an entity co-occurrence graph (constructed using lightweight NLP tools instead of LLMs). The model builds bidirectional indexes between entities and chunks to enable adaptive retrieval that automatically switches between local and global query modes. Experiments on long-document QA datasets show that E2GraphRAG achieves up to 10× faster indexing and 100× faster retrieval than GraphRAG, while maintaining comparable QA accuracy.

**Strengths:**

Well-motivated improvement over GraphRAG, targeting efficiency–effectiveness trade-offs.

Clever use of non-LLM entity extraction tools to reduce computational cost.

Comprehensive experiments and ablation studies that clearly demonstrate both speed and accuracy gains.

**Weaknesses:**

The paper’s novelty lies mainly in the integration and efficiency-oriented redesign of existing graph-based RAG components rather than in proposing a fundamentally new retrieval or reasoning paradigm. It offers practical, well-engineered improvements but limited conceptual originality.

**Questions:**

N/A.

---

> ### Author Response · Authors · 2025-11-14
>
> We highly appreciate your understanding of our paper, especially the novelty part. Thank you again for your positive comment.

---

### Official Review · Reviewer_ties · 2025-11-01

**Soundness:** 2
**Presentation:** 2
**Contribution:** 2
**Rating:** 2
**Confidence:** 4

**Summary:**

This paper introduce $E^2$GraphRAG, a GraphRAG method with NLP tools to construct graphs. Specifically, the authors first build a hierarchical chunk summarization tree, and then enrich it by linking chunk nodes to an entity-level knowledge graph extracted using NLP techniques. For each query, the proposed retrieval strategy adaptively selects either local or global retrieval. Experimental results show that the method outperforms existing baselines while maintaining efficiency.

**Strengths:**

1. The designed retrieval mechanisim can automatically choose local or global retrieve.
2. The authors compared the number of input and output tokens of different methods.
3. The authors provided detailed ablation study of the proposed method to demonstrate the effectiveness of different designs.

**Weaknesses:**

1. The novelty of the proposed method is limited. Constructing knowledge graphs using NLP tools such as entity and relation extraction has been widely explored in prior NLP and RAG-related studies, and thus cannot be considered a key contribution of this work. In addition, the hierarchical summarization tree is largely inspired by RAPTOR, which further reduces the originality of the approach.

2. The overall writing quality needs improvement. The paper reads more like a technical report rather than a well-structured academic paper. It lacks clear motivation, and the rationale behind each design choice is not well explained. Many components appear ad hoc, without sufficient theoretical grounding or empirical justification.

3. The experiments are only conducted on three datasets, which is not sufficient to demonstrate the robustness and generalizability of the proposed method. Additional benchmarks, especially multi-hop QA datasets such as MultihopRAG, should be included to more comprehensively validate the effectiveness of the approach.

**Questions:**

1. In the retrieval, "queries whose entities are densely connected are processed locally, while others fall back to global retrieval." What if there is only one entity in the query?

2. What is the reason that the authors summary the consecutive chunks? What if chunks in different documents share the similar semantic meaning?

3. Where did the authors get the heuristic of Graph Filtering, "truly relevant entities tend to be semantically related and thus connected in the constructed graph".

---

> ### Author Response · Authors · 2025-11-14
> **Response by Authors [1/2]**
>
> Thank you for your questions and suggestions. Here are our responses.
>
> **W1. Limited Novelty**
>
> We respectfully refer you to our global response, where we provide a detailed clarification of our contributions and the concerns related to novelty. **While constructing knowledge graphs using NLP tools such as entity and relation extraction has been widely explored in prior NLP studies, it has not been introduced in the study of GraphRAG**. Prior GraphRAG methods are all based on LLMs.
>
> **W2. Low Presenting Quality**
>
> Thank you for your suggestions, we break this point into three sub-questions and response them consequently.
>
> *W2.1 Lack of Clear Motivation*
>
> As stated in line 60-68, our motivation comes from the existing challenges. To address the efficiency problem, we try to leverage the lightweight NLP methods to replace the LLM for NER and relation extraction. However, the entity graph only cannot support the global query, we further combine the entity graph and the summary tree to support both local and global query, which is corresponding to the second challenge mentioned in the paper (line 62-64). After indexing, we leverage the entity graph as a filter, and design an adaptive retrieval mechanism, which solves the third challenge.
>
> *W2.2 Lack of Rationale behind Each Design Choice*
>
> We believe that the rationale is well explained in our script. For each component, we list the corresponding rationale in our paper.
>
> 1. Graph Filter:
>    line 240-243. *...it often fails to identify the core entities aligned with the query intent...*
> 2. Entity-Aware Ranking:
>    line 279-281. *If the latter occurs (i.e., the retrieval result becomes empty)...* explains why we need this mechanism.
>    line 283-285. *Chunks covering more entites ... tend to contain more comprehensive contextual information* explains why we use the Entity Coverage Ranking.
> 3. Global Dense Retrieval:
>    line 234-235. *... we cannot utilize the entities to support meaningful retrieval...treat as a global query.* explains why we need the Global Dense Retrieval.
> 4. Occurrence Ranking:
>    line 257-259. *... entities related to both question and context ... can assist in improving chunk selection.* explains why we conduct occurrence ranking.
>    line 263-264. *This recursive weighting naturally assigns higher scores to high-level summary nodes, aligning with the intuition behind global retrieval.* explains the intuition behind the credit assignment.
>
> If any further clarification is needed, please kindly indicate specifically.
>
> *W2.3 Lack of Empirical Justification for Components*
>
> We kindly refer you to check the Table 3 in our paper, which reports the results of ablation study for almost each component. The results show that removing any component leads to a degraded performance, which provides strong empirical justification. You also mention that we have provided detailed ablation study of the proposed method as one of our strengths.
>
> **W3. Lack of Experiments on Multi-hop QA Datasets**
>
> Thank you for your suggestion. We conduct additional experiments on the HotpotQA dataset (randomly sampling 1,000 entries from the validation set) and a **domain-specific dataset** from GraphRAG-Bench [1], namely Medical. All baseline methods share the same backbone model, **Llama3-8B**, to ensure fair comparison. For HotpotQA, we report **Contain-Match** (HotpotQA-Contain), and **GPT-Evaluation Accuracy** (HotpotQA-GPT), while for medical, we use **GPT-Evaluation Accuracy** (Medical-GPT) only, as this dataset poses greater challenges for automatic evaluation.
>
> The experimental results are summarized below:
>
> |      | HotpotQA-Contain | HotpotQA-GPT | Medical-GPT |
> |------|------|------|------|
> | RAPTOR | 55.9  | 58.3 | 55.75 |
> | LightRAG | 60.3  | 59.5 | 54.36 |
> | HippoRAG | 57.0 | 59.3 | 55.04 |
> | E$^2$GraphRAG | **61.0** | **63.9** | **58.00** |
>
> The results show that, although our method does not include any specific design for multi-hop questions, it still achieves **top-tier performance**. We attribute this to the **inherent multi-hop reasoning capability** introduced by the entity graph structure.
>
> For efficiency, we additionally report the indexing time and retrieval time on multi-hop datasets. These metrics reflect the end-to-end system cost and allow a fair comparison across different graph-based RAG frameworks. The results are summarized in the table below.
>
> |      | Indexing Time | Retrieval Time |
> |------|------|------|
> | RAPTOR | 1323.57  | 0.062 |
> | LightRAG | 4933.22  | 10.963 |
> | HippoRAG | 936.00 | 1.461 |
> | E$^2$GraphRAG | **534.60** | **0.053** |
>
> [1] Xiang et al., When to use Graphs in RAG: A Comprehensive Analysis for Graph Retrieval-Augmented Generation

---

> ### Author Response · Authors · 2025-11-14
> **Response by Authors [2/2]**
>
> **Q1. How to manage the situation that only one entity in the query**
>
> As stated in line 255-257, *After this filtering step, if no entity pairs meet the criteria, ...,we classify it as a coarse-grained global query as well. **This also includes cases where the query contains only a single entity, as there are no pair-wise combinations.***
>
> **Q2. What if chunks in different documents share the similar semantic meaning**
>
> They will be connected by our entity graph, which is stated in line 209-210. *Since some entities appear in multiple chunks, this merging allows the graph to capture the co-occurrence relationships among entities across the entire document.*
>
> **Q3. Why relevant entities tend to be connected in the constructed graph**
>
> We can justify this conclusion based on a widely adopted assumption and the rules we used to construct the entity graph. The first assumption is that the meaning of a word depends on its surrounding context, which underlies models such as BERT, GloVe, and other word embeddings. Our graph construction rule connects only entities that appear in the same sentence, meaning they share the same surrounding context and are therefore more likely to be semantically related.
>
> During the retrieval stage, we do not limit the hop distance to 1, allowing the model to discover multi-hop entity relationships. In other words, the relations are not confined to a single sentence, but can extend over longer contexts, enabling richer reasoning across multiple entities.

---

### Official Review · Reviewer_rZe1 · 2025-11-01

**Soundness:** 2
**Presentation:** 3
**Contribution:** 2
**Rating:** 4
**Confidence:** 4

**Summary:**

This paper introduces E2GraphRAG, a novel Retrieval-Augmented Generation framework that advances the Pareto frontier of efficiency and effectiveness in graph based RAG systems. Its core innovation lies in combining a hierarchical summary tree built with large language models and an efficient entity co occurrence graph constructed using traditional NLP tools like SpaCy. These two components are interconnected through bidirectional entity to chunk and chunk to entity indexes. This hybrid structure supports adaptive retrieval by allowing queries to dynamically trigger either local entity centric retrieval or global dense embedding retrieval, based on detected query characteristics and graph structure. This approach eliminates the rigidity and computational inefficiency found in prior work such as GraphRAG.

**Strengths:**

- The writing and figures of the paper are clear.
- The discussion of indexing efficiency in graph-based RAG systems for corpora with long documents is meaningful.

**Weaknesses:**

- The main concern is the novelty in the proposed framework. The core module for constructing the graph combines the "merge and summarize text chunks to construct a hierarchical tree" approach (similar to RAPTOR) with building a concise entity graph from dispersed chunks (similar to knowledge graph construction-based methods, such as GraphRAG and HippoRAG2). The use of lightweight tools such as SpaCy and BERT is also not new compared to existing methods. As a result, the framework seems to be an incremental integration of existing baselines, rather than offering strong innovation.
- In the experiments, the base LLM used in this paper is not consistent with the baselines. Key baselines, such as RAPTOR, HippoRAG2, and LightRAG, are evaluated using stronger generation models, including LLaMA-3.3-70B-Instruct and GPT-4o-mini API, whereas all models are tested solely with a 7B-scale LLM in this paper. This mismatch raises concerns about the credibility of the experimental results, particularly since the proposed method claims to offer better accuracy and efficiency.
- The experiments only consider datasets with long documents, ignoring classic multi-hop question answering GraphRAG datasets [1] such as MuSiQue, 2WikiMultihopQA, and HotpotQA. This omission may lead to unfair comparisons. While indexing efficiency and accuracy on long documents are important, evaluations on standard multi-hop datasets should not be neglected.

[1] From RAG to Memory: Non-Parametric Continual Learning for Large Language Models

**Questions:**

As noted in the Weaknesses section.

---

> ### Author Response · Authors · 2025-11-14
>
> Thank you for your questions and suggestions. Here are our responses.
>
> **W1. Lack of Novelty**
>
> We respectfully refer you to our global response, where we provide a detailed clarification of our contributions and the concerns related to novelty.
>
> **W2. Lack of Experiments on Stronger Models**
>
> Thank you for the suggestion. For fair comparison, we conduct experiments under the same experimental setup as HippoRAG2 and directly report results of SOTA methods as provided in their original paper. We show the Exact Match (EM) and token cost results based on the stronger backbone model Llama3.3-70B-Instruct on HotpotQA (1000 samples from validation set) as below.
>
> |      | EM | Input Token | Output Token |
> |------|------|------|------|
> | RAPTOR | 56.8  | **1.7M** | **0.2M** |
> | GraphRAG | 55.2  | 115.5M | 36.1M |
> | HippoRAG | 52.6 | 9.2M | 3.0M |
> | HippoRAG2 | **62.7** | 9.2M | 3.0M |
> | E$^2$GraphRAG | 61.3 | **1.7M** | **0.2M** |
>
> We find that based on the stronger model, our method still performs well, only slightly lower than the HippoRAG2, which may be triggered by random fluctuations caused by the small sample size. In addition, we emphasize that our method and RAPTOR tie for the lowest computational cost and token consumption among all compared methods, further highlighting our performance advantage.
>
> In addition to these extra experiments, we would like to further clarify our motivation. As stated in Lines 308–310, our framework is primarily designed for practical scenarios under **limited-resource and privacy-constrained environments**, where larger models are infeasible due to computational limitations, and closed-source models cannot be adopted due to privacy restrictions.
>
> **W3. Lack of Experiments on Multi-hop QA Datasets**
>
> Thank you for your suggestion. We conduct additional experiments on the HotpotQA dataset (randomly sampling 1,000 entries from the validation set) and a **domain-specific dataset** from GraphRAG-Bench [1], namely Medical. All baseline methods share the same backbone model, **Llama3-8B**, to ensure fair comparison. For HotpotQA, we report **Contain-Match** (HotpotQA-Contain), and **GPT-Evaluation Accuracy** (HotpotQA-GPT), while for medical, we use **GPT-Evaluation Accuracy** (Medical-GPT) only, as this dataset poses greater challenges for automatic evaluation.
>
> The experimental results are summarized below:
>
> |      | HotpotQA-Contain | HotpotQA-GPT | Medical-GPT |
> |------|------|------|------|
> | RAPTOR | 55.9  | 58.3 | 55.75 |
> | LightRAG | 60.3  | 59.5 | 54.36 |
> | HippoRAG | 57.0 | 59.3 | 55.04 |
> | E$^2$GraphRAG | **61.0** | **63.9** | **58.00** |
>
> The results show that, although our method does not include any specific design for multi-hop questions, it still achieves **top-tier performance**. We attribute this to the **inherent multi-hop reasoning capability** introduced by the entity graph structure.
>
> For efficiency, we additionally report the indexing time and retrieval time on multi-hop datasets. These metrics reflect the end-to-end system cost and allow a fair comparison across different graph-based RAG frameworks. The results are summarized in the table below.
>
> |      | Indexing Time | Retrieval Time |
> |------|------|------|
> | RAPTOR | 1323.57  | 0.062 |
> | LightRAG | 4933.22  | 10.963 |
> | HippoRAG | 936.00 | 1.461 |
> | E$^2$GraphRAG | **534.60** | **0.053** |
>
> [1] Xiang et al., When to use Graphs in RAG: A Comprehensive Analysis for Graph Retrieval-Augmented Generation

---

### Author Response · Authors · 2025-11-14
**Global Response to Concerns Regarding Novelty**

We sincerely thank all the reviewers for their feedbacks. Regarding the novelty issue, we respectfully disagree with the comment on the lack of novelty of our paper.

On the one hand, prior to our work, all existing GraphRAG approaches are too **heavy**, which rely on LLMs for entity and relation extraction, and introduce significant computational and financial costs. Therefore, despite the effectiveness, they are hard to widely use. To address the problem, we propose the first non-llm-based lightwight graph construction method for GraphRAG, which achieves comparable or even better effectiveness as LLM-based models with significant speedup in both indexing and retrieval stages. This is the core contribution of our paper. **In particular, we never claim that the use of lightweight tools is our contribution (as misunderstood by Reviewers rZe1 and ties), but we are the first to point out the new direction that GraphRAG without LLMs for graph construction and retrieval can still perform well**. This paves the way for the wide deployment of GraphRAG and also sheds light on the direction for feasible future study. ``Towards this direction, if small models can be used in GraphRAG to offer practical improvements, why do we need to use costly LLMs?``

On the other hand, for the retrieval stage, we have our own characteristic. Existing approaches mainly lie in two ways. First, GraphRAG and LightRAG treat the knowledge graph as the source of the information. Second, HippoRAG and HippoRAG2 treat it as an auxiliary component to enhance the performance of embedding based retrieval. Different from them, our method leverages the entity graph as a filter to identify the core entities in the query, which further help locate the relevant chunks. **Note that our constructed graph is an entity graph, but not a knowledge graph**. This also highlights the difference between our method and other baselines in the indexing stage.

In summary, we provide insights in both indexing stage and retrieval stage. Therefore, we believe our contribution goes beyond an incremental improvement or technical report, marking an initial step toward a new research direction within the GraphRAG framework.

---

### Author Response · Authors · 2025-11-27

Dear Reviewers,

We first want to show our most sincere appreciations for you providing us with insightful comments. We have clarified the novelty part (which has been acknowledged by Reviewer XF2c) and also conduct additional experiments as suggested. We totally understand how busy you are, while we also hope our efforts can be recognized. Hence, could you please take a look at our responses and give us some feedbacks? Thank you.

Best,

Authors

---

### Author Response · Authors · 2025-12-02
**Global Summary for Area Chair and Reviewers [1/2]**

Dear Area Chair and Reviewers,

We thank you again for your efforts toward fair and rigorous reviewing. To help you quickly understand our paper and the issues raised during the rebuttal, we provide the following summary.

**Paper Summary**

This paper introduces E$^2$GraphRAG, the first GraphRAG-style method that replaces LLM-based named-entity extraction with traditional NLP tools (e.g., SpaCy, BERT) during graph construction. This design preserves QA accuracy while substantially improving efficiency, pushing the Pareto frontier of GraphRAG methods.

**Main Concerns**

1. Novelty of the Proposed Method

The novelty appears in both the graph construction and retrieval stages. Prior GraphRAG methods rely on LLMs for graph construction, resulting in prohibitive cost and long indexing time, which hinder practical deployment. To address this, we propose an LLM-free entity graph construction approach that significantly reduces cost while maintaining competitive QA performance. Note that our graph is an entity graph, not a knowledge graph, which differentiates our indexing procedure from existing baselines.

In retrieval, prior GraphRAG methods generally follow two paradigms:
(1) relying entirely on the graph for information (e.g., GraphRAG, LightRAG), or
(2) using the graph to assist chunk retrieval (e.g., HippoRAG, HippoRAG2).
In contrast, our method uses the entity graph as a filter to identify core query entities, which then guide the chunk selection process. Further details are provided in Global Response to Concerns Regarding Novelty.

2. Experimental Coverage

During the rebuttal, we added experiments on multihop QA tasks, evaluated stronger base models, and included additional domain-specific datasets. Across all settings—different tasks, model scales, and domains—our method remained effective and consistently achieved the highest graph construction efficiency. Detailed results are provided in the Global Summary for Area Chair and Reviewers [2/2].

3. Efficiency Improvements

Regarding Reviewer Zxuz’s concerns, we offer the following clarification:

Our method does not drastically reduce input tokens, as we intentionally preserve the “global query” capability emphasized in the GraphRAG literature. If only local responses were required, we could omit the document summary tree entirely. This would reduce indexing time and token count to levels comparable to contemporaneous methods such as LinearRAG or NoLLMRAG. However, doing so would deviate from our goal of improving the GraphRAG family rather than redefining its scope.

Despite maintaining global-query capability, our method still achieves efficiency gains. Even compared to RAPTOR—the most efficient baseline—Figure 5 shows that our method exhibits the lowest slope as the corpus scales, meaning our processing time grows linearly and at the slowest rate. At the same time, our model consistently outperforms RAPTOR in QA accuracy. This demonstrates improvements on both efficiency and effectiveness, rather than an incremental refinement.

We once again thank you for your efforts toward maintaining academic fairness and helping improve our work. We sincerely hope that your final assessment will reflect your expertise and objective judgment.

---

### Author Response · Authors · 2025-12-02
**Global Summary for Area Chair and Reviewers [2/2]**

Here we summarize our supplement experiments.

**Experiments on Larger Language Models**

For fair comparison, we conduct experiments under the same experimental setup as HippoRAG2 and directly report results of SOTA methods as provided in their original paper. We show the Exact Match (EM) and token cost results based on the stronger backbone model Llama3.3-70B-Instruct on HotpotQA (1000 samples from validation set) as below.

|      | EM | Input Token | Output Token |
|------|------|------|------|
| RAPTOR | 56.8  | **1.7M** | **0.2M** |
| GraphRAG | 55.2  | 115.5M | 36.1M |
| HippoRAG | 52.6 | 9.2M | 3.0M |
| HippoRAG2 | **62.7** | 9.2M | 3.0M |
| E$^2$GraphRAG | 61.3 | **1.7M** | **0.2M** |

We find that based on the stronger model, our method still performs well, only slightly lower than the HippoRAG2, which may be triggered by random fluctuations caused by the small sample size. In addition, we emphasize that our method and RAPTOR tie for the lowest computational cost and token consumption among all compared methods, further highlighting our performance advantage.

In addition to these extra experiments, we would like to further clarify our motivation. As stated in Lines 308–310, our framework is primarily designed for practical scenarios under **limited-resource and privacy-constrained environments**, where larger models are infeasible due to computational limitations, and closed-source models cannot be adopted due to privacy restrictions.

**Experiments on Multi-hop QA Datasets and Domain-specific Dataset**

We conduct additional experiments on the HotpotQA dataset (randomly sampling 1,000 entries from the validation set) and a **domain-specific dataset** from GraphRAG-Bench [1], namely Medical. All baseline methods share the same backbone model, **Llama3-8B**, to ensure fair comparison. For HotpotQA, we report **Contain-Match** (HotpotQA-Contain), and **GPT-Evaluation Accuracy** (HotpotQA-GPT), while for medical, we use **GPT-Evaluation Accuracy** (Medical-GPT) only, as this dataset poses greater challenges for automatic evaluation.

The experimental results are summarized below:

|      | HotpotQA-Contain | HotpotQA-GPT | Medical-GPT |
|------|------|------|------|
| RAPTOR | 55.9  | 58.3 | 55.75 |
| LightRAG | 60.3  | 59.5 | 54.36 |
| HippoRAG | 57.0 | 59.3 | 55.04 |
| E$^2$GraphRAG | **61.0** | **63.9** | **58.00** |

The results show that, although our method does not include any specific design for multi-hop questions, it still achieves **top-tier performance**. We attribute this to the **inherent multi-hop reasoning capability** introduced by the entity graph structure.

For efficiency, we additionally report the indexing time and retrieval time on multi-hop datasets. These metrics reflect the end-to-end system cost and allow a fair comparison across different graph-based RAG frameworks. The results are summarized in the table below.

|      | Indexing Time | Retrieval Time |
|------|------|------|
| RAPTOR | 1323.57  | 0.062 |
| LightRAG | 4933.22  | 10.963 |
| HippoRAG | 936.00 | 1.461 |
| E$^2$GraphRAG | **534.60** | **0.053** |

[1] Xiang et al., When to use Graphs in RAG: A Comprehensive Analysis for Graph Retrieval-Augmented Generation

---

### Meta-Review · Area_Chair_SQiy · 2025-12-04

**Summary:**

This paper introduce a GraphRAG method with NLP tools to construct graphs. Specifically, the authors first build a hierarchical chunk summarization tree, and then enrich it by linking chunk nodes to an entity-level knowledge graph extracted using NLP techniques. For each query, the proposed retrieval strategy adaptively selects either local or global retrieval. Experimental results show that the method outperforms existing baselines while maintaining efficiency.

**Reviewer Concerns:**

Most reviewers think it may be below the boardline. Their main concerns are on novelty and experiments. I think these concerns are not addressed by rebuttals.

**Reviewer Scores:**

I do not think the reviewers will change their scores.

---

### Decision · Program_Chairs · 2026-01-26

Reject